# A Multicriteria Standard to Rank Plea Bargain Proposals

Annibal Parracho Sant'Anna [1,*,†], Luiz Octávio Gavião [2,†] and Tiago Lezan Sant'Anna [3,†]

[1] Programa de Doutorado em Sistemas de Gestão Sustentáveis, Universidade Federal Fluminense, Rua Passo da Pátria, 156, Niteroi 24210-240, RJ, Brazil

[2] Programa de Pós-Graduação em Segurança Internacional e Defesa, Escola Superior de Guerra, Av. João Luiz Alves, s/n, Rio de Janeiro 22291-090, RJ, Brazil

[3] Banco Nacional de Desenvolvimento Econômico e Social, Av. Chile, 100, Rio de Janeiro 20031-917, RJ, Brazil

[*] Correspondence: annibal.parracho@gmail.com or aparracho@id.uff.br

[†] These authors contributed equally to this work.

**Abstract:** This article presents a model for the comparison of plea bargain proposals. The use of the model increases the possibility of the satisfactory development of the negotiation of rewarded collaboration agreements recently permitted under Brazilian law. A novelty in the model is the objective consideration of society's interest in adequately punishing defendants whose guilt can be proven. To allow for the inclusion of this element, a multicriteria approach that adds the criteria representing the prosecution's aims to the criteria regarding the accused's positions is adopted. The importance of the criteria is derived without direct criteria weighting. A novel joint treatment to criteria collinearity and interaction is developed, which enables the model to accommodate any number of defendants, proposals, and criteria. The framework so developed enhances transparency and encourages collaboration. By assigning a new meaning to the plea bargain, it is able to bring about the necessary shift in cultural standards that can lead to the effective weakening of criminal organizations.

**Keywords:** plea bargain; rewarded collaboration; criminal organization; composition of probabilistic preferences; interaction; collinearity

## 1. Introduction

Since [1], lawsuits have been considered within the broad context of negotiation. Efficient methods have been developed to evaluate the convenience of judicial determinations by comparing, in economic terms, each party's costs and benefits. With [2,3], different methodologies for evaluating negotiations that are substituted for the penal process started being developed.

In negotiations involving criminal law, political factors may become important [4]. In fact, one of the parties in the criminal process, the prosecution, assumes the responsibility of representing the values of society beyond the sphere of citizen welfare. The formulation of negotiations in a comprehensive framework helps limit the possibility of public agents in a co-ordinated hierarchy [5] personally exercising power that belongs to institutions.

The central issue is that this institutional pertinence faces obstacles derived from power ambitions that are especially relevant when combating violence [6]. To ensure their prevalence, it is useful to be able to precisely determine in the agreement framework how societal values, above the interests of the individuals in the principal-agent chain, are implemented.

In the last decades, this aspect of the economic analysis of the criminal justice systems has presented important advances [7,8]. A line of thought, represented, for instance, by [9], investigates how institutional cost bounds may limit the harshness of penal rules. Other investigations, such as [10], study the effects of the difficulty for defendants in facing the costs of the criminal justice system. The implications of different such models for the design of criminal procedure rules are reviewed in [11].

The role of the prosecutor has been a central issue of these analyses [12–14]. In these developments, the objectives vary, from the reduction of the cost of the judicial procedures and of the expenditures with the care of the inmates [15,16] to the efficiency in fighting crime [17,18] and avoiding the conviction of innocents [19,20].

In practice, what is more relevant is the effort to develop out-of-court solutions. In the United States, this happened through the overwhelming use of the plea bargain [21,22]. The efficiency of the system developed by exploring the plea bargain, both to the reduction of crime and incarceration rates and to the production of justice, has been disputed.

A similar resource to help improve the negotiation procedures preceding the installation of the process has been established in Brazil: the rewarded collaboration mechanism, established by Law 12850/2013 (Law on Organized Crime) and additionally regulated by Law 13964/2019 (Anti-crime Law).

The possibility of rewarded collaboration may lead misdemeanor-indicted defendants to denounce big crime bosses. Considering multiple criteria and different points of view, a probabilistic comparison of collaboration proposals as a tool in the rewarded collaboration system may lead it to be perceived as more aligned to societal values. This may cause it to be more easily acceptable by possible collaborators.

The justice system in Brazil faces overcrowding in prisons, mostly with poor people led to becoming involved in selling small amounts of drugs, along with low productivity in completing the prosecution of more serious criminal cases [23]. Although Brazil is third in terms of total imprisonment, with fewer people in jail than the United States and Russia, the phenomenon of mass incarceration [24] is more unbearable in Brazil due to the rate of almost two persons per each available prison spot. In this context, rewarded collaboration gains relevance.

The possibility of reducing a defendant's penalty in exchange for information has been present in Brazilian law for a long time. The most noticeable instance is the provision in Law 8072/1990 (Law on Hideous Crimes) for a sentence reduction of 1/3 to 2/3 for a collaborator who reports a kidnapping to the authorities, facilitating the release of the kidnapped individual. Sentence reductions of the same size are also offered to collaborators in Law 7492/1986 (Law on White-Collar Crime) and Law 9613/1998 (Law on Money Laundering). In the sequence, Law 9807/1999 (Law on Witness Protection) authorizes granting a judicial pardon and the consequent extinction of the punishment to the accused who, as a first offender, collaborated effectively and voluntarily with the investigation and the criminal prosecution. However, the regulation of the systematic use of sentence reductions in a detailed negotiation procedure came only recently, with the law on organized crime and the Anti-crime Law.

In Brazilian law, rewarded collaboration is essentially a negotiation leading to an agreement regarding information provided by a defendant in exchange for a reduced sentence. The prosecution presents the result of the negotiation to a judge, who either accepts the defendant's testimony and approves the reduced sentence or rejects both. The prosecution will eventually propose criminal prosecution for suspects who refuse to negotiate or who do not provide information deemed useful enough to justify a reduced sentence.

The Anti-crime Law defines rewarded collaboration as a negotiation within the legal process and a means of acquiring evidence. This implies, on the one hand, that it is incorporated into the criminal process and, on the other hand, that it does not need to include legal evidence, but rather is only a means of obtaining evidence. The defendant in a negotiation must explain how to locate evidence but is not required to provide it directly. The defendant is responsible for developing the collaboration proposal and attaching a proper description of the facts, including all circumstances, criminal evidence, and corroboration elements.

Rewarded collaboration has an ample scope. To be accepted, it must result in one of the following: (I) the identification of the other coconspirators and participants in the criminal organization and the criminal offences committed by them; (II) the disclosure of the hierarchical structure and division of tasks within the criminal organization; (III) the

prevention of criminal offences arising from the activities of the criminal organization; (IV) the total or partial recovery of the product or benefit of the criminal offences committed by the criminal organization; and (V) the details of the present location of any possible victims, with their physical integrity preserved.

The receipt of a formal proposal of collaboration marks the beginning of a negotiation. It is also a sign of confidentiality; until the final lifting of secrecy by a judicial decision when a criminal complaint is formally received, the disclosure of the initial negotiation or any document formalizing it constitutes a breach of confidentiality and a breach of trust and good faith. In the event that a collaboration agreement is not signed by the authority, the latter is not permitted to use any of the information or evidence presented by the collaborator for any other purpose.

Negotiations and acts of collaboration must be recorded, and a copy of the records must be made available to the collaborators. Furthermore, a collaborator's attorney, in the interests of exercising the collaborator's right to a defence, has ample access to the process, with the exception of evidence related to ongoing investigative proceedings.

The revision of the concept by the Anti-crime Law considerably expanded the protection of collaborators. Thus, it offers them the right to serve their sentences in a prison different from that of their group's other members. In addition, collaborators have the right to appear in court separately from other participants and without eye contact with other suspects. In addition, no agreement should be finalized without the presence of the collaborator's lawyer or public defender.

Even so, the mistrust of a defendant with a lower social status towards people of a higher status is not eliminated. It is expected that a more institutionalized environment may provide defendants with a greater sense of personal safety.

Rewarded collaboration may become an effective tool in the fight against criminal organizations if it is able to attract more defendants to a negotiated conflict resolution. These defendants may offer information that is otherwise very difficult to obtain. However, implicit violence within the system scares potential collaborators.

Initially, rewarded collaboration under the law on organized crime was effectively applied only in the context of white-collar crime, at its highest levels, where important company managers could count on the assistance of the best lawyers to manage their collaborations. Its application at the top of bribery chains was useful, subjecting political elites to the law.

Nevertheless, it was not used on a large scale. Poorer defendants, whose contributions would certainly be useful in prosecuting violent gangs, which might be more important to the population at large, do not trust the authorities. The risk of having a proposal rejected and being sent to prison without protection against the rest of the gang now in jail is a serious danger.

The availability of an attractive tool for the competition between codefendants that involves the opportunity to collaborate with the justice system may be the source of the desired cultural change. With an objective system to evaluate collaboration proposals, limiting the authorities' intervention, and making the process more transparent, there is reason to expect a change in the defendants' willingness to rely on state protection.

Negotiation is more realistic when the possible variations in penalties are considered together with the associated workload. This leads to a combination of criteria measuring the satisfaction or dissatisfaction not only of the suspects with the penalties assigned but also of the prosecution and of society with the abbreviation of the criminal investigation. This poses a dilemma for the authorities, who are pressured by the opposition between their duty to society to punish criminals and the costs involved in complying with this duty.

In [25], a model to address the problem of classifying rewarded collaboration proposals, as defined in the Anti-crime Law, is developed. Each proposal is evaluated separately and allocated in one item from an ordered list of predetermined classes. Here, instead of the allocation of proposals in previously determined classes, the final output is a vector of scores ranking all the alternatives.

The problem is then modeled in terms of a multicriteria negotiation [26] in which a number of alternatives are compared.

The alternatives are determined based on the penalties that may apply to the defendants according to their collaboration proposals and on variables measuring the costs and benefits to the prosecution in terms of the limits of the information obtained in exchange for each set of penalties.

One prosecution variable is the cost of continuing an investigation to be able to decide whether to accept a collaboration agreement. Another criterion considered is the relevance of the information to the dismantling of criminal organizations. Reducing the power of such organizations is the most important goal for prosecutors.

The alternatives are compared based on the probabilities of minimizing the values of the attributes considered, i.e., penalties, costs, criminal organizations' power, and any other evaluations of the alternatives according to the available criteria. A multicriteria decision rule is employed, which considers society's wishes behind those of the defendants and prosecution regarding reducing penalties and collecting information. Alternatives chosen by applying a society-oriented rule are more likely to be accepted by all.

To build this rule, a single principle is initially followed: the principle of concentration of preferences [27]. This principle seeks to take into account the uncertainty that may alter the evaluations [28]. A basic assumption of decision analysis is the existence of a best alternative. This leads to valuing any criterion by its ability to point to a best alternative. Based on this idea, the principle of concentration of preferences gives a greater importance to those criteria or sets of criteria that most clearly highlight an alternative as the best in the set of alternatives considered.

To consider also the possibility of interaction between criteria, the composition rule uses the Choquet integral [29,30]. The principle of concentration of preferences is used to derive from the matrix of probabilities of preference a Choquet capacity in relation to which the integration is performed. An interaction between criteria may occur, for instance, due to communication between defendants.

The initial procedure of the composition of probabilistic preferences (CPP) [31,32] is used to form the preference probability matrix employed. In it, the principle of concentration of preferences is applied in a probabilistic transformation that widens the distance between preferred alternatives. In addition, the feature of the Choquet integral that assigns a greater importance to high evaluations according to criteria with higher positive interactions aligns with the same principle.

A new component in this procedure supplements the standard established in [33]. It is intended to facilitate the incorporation of a large number of criteria and alternatives. It consists of a mechanism that considers the correlation between the vector of evaluations by each criterion with the vectors of evaluations by other criteria to reduce its weight.

This paper is organized in the following manner: After this introduction of the problem, Section 2 details the multicriteria approach developed. In Section 3, the evaluation system is described. An example of a numerical application is then presented. Section 4 concludes the study.

## 2. Materials and Methods

The ranking methodology developed here is based on the use of CPP to combine the evaluations by multiple criteria. CPP relies on an initial transformation of each criterion's evaluations into probabilities of being the best. The estimation of these probabilities is here performed using a straightforward counting procedure that disregards exclusively cardinal variations. The probabilistic transformation makes possible the use of the Choquet integral to account for interaction when combining the criteria. A discount rule to reduce the influence of collinearity between criteria is included.

*2.1. The Composition of Probabilistic Preferences*

CPP is a methodology that considers, in the composition of multiple criteria, the probabilistic character of preference assessment. This probabilistic character may always be present, resulting, for instance, from the imprecision caused by subjective factors that lead decision makers to attribute different meanings to the same attributes of alternatives in different circumstances or simply from measurement errors that affect the evaluations of such attributes.

A fundamental step of CPP is the transformation of the vector of the evaluations of different alternatives according to each criterion into a vector of probabilities of each alternative being the preferred one. This can be performed as follows:

Let $(a_{1j}, \ldots, a_{nj})$ be the vector of numerical evaluations of n alternatives $A_1, \ldots, A_n$ by the criterion $C_j$. For each k, from 1 to n, and j, from 1 to m, the size of the set S of criteria, let $X_{kj}$ denote a random variable with the distribution of preference for alternative $A_k$ according to criterion $C_j$. In the absence of more accurate information, $X_{kj}$ will have preference probabilities directly derived from the values of the trichotomic pairwise comparisons between the alternatives.

Let's denote by $C_{ij}$ the count of preferences for the i-th alternative according to $C_j$ and by $A(j,i_1,i_2)$ the result of the trichotomic comparison between alternatives $i_1$ and $i_2$ according to the j-th criterion. To $A(j,i_1,i_2)$ is assigned one of three possible values: 1 if $i_1$ is preferable to $i_2$ by $C_j$, 0 if $i_2$ is preferable to $i_1$, and $1/2$ if there is indifference between $i_1$ and $i_2$ by $C_j$.

The preference count for the i-th alternative according to the j-th criterion is given by the sum

$$C_{ij} = \sum_{i2} A(j,i,i_2) \tag{1}$$

for $i_2$ ranging over all the n-1 alternatives $i_2$ that $A_i$ is compared with.

The preference for $A_i$ is therefore the sum of the number of pairwise comparisons where i is preferred with half the number of comparisons where $A_i$ is considered equivalent to another alternative. The estimate of the probability of preference for $A_i$ by $C_j$ is the quotient

$$P_{ij} = C_{ij}/(n(n-1)/2) \tag{2}$$

of the count $C_{ij}$ by the number of comparisons.

The sum of these probabilities along the set of alternatives is exactly 1.

It is interesting to note that, granting antisymmetry,

$$A(j,i_1,i_2) = 1 - A(j,i_2,i_1). \tag{3}$$

*2.2. The Treatment of Collinearity*

In the present study, a novel device is created to address the possibility of collinearity. We may need to limit the impact on the counting of the presence of a same factor affecting two or more criteria. Collinearity has nothing to do with the effect of interaction between criteria that actually amplifies or reduces their individual effects, but only with the effect on the measurements produced by the presence of common components in the criteria.

The counting approach enables us to extend from only one criterion to a subset J of the set of criteria, determining a preference score $P_{iJ}$ for alternative $A_i$ according to J by adding the $C_{ij}$ along the criteria j in J. However, the simple addition of the counts of the preferences by the elements of J may be distorted by the presence of common factors implicit in different criteria, what would lead us to overvalue such criteria.

Different criteria equally ranking all the alternatives must not be present. In addition, to account for the influence of common factors in different criteria and for the possibility of interaction between the criteria, the preference according to sets of criteria is measured by a capacity. In the construction of this capacity, to account for collinearity of criteria whose vectors of preference exhibit positive rank correlation, a new rule to share common factors is here designed.

Denoting by $S_{j1j2}$ the Spearman correlation coefficient between vectors of evaluations $(a_{1j1}, \dots, a_{nj1})$ and $(a_{1j2}, \dots, a_{nj2})$, for $S_{j1j2} > 0$, a portion of the $P_{ij1}$ and $P_{ij2}$ proportional to $S_{j1j2}$ is subtracted from each of them. For $j_1$ and $j_2$ with positive $S_{j1j2}$ and without positive correlation with other criteria, this leads to a reduction of their individual importance by subtracting $P_{ij1}S_{j1j2/2}/2$ from $P_{ij1}$ and $P_{ij2}S_{j1j2/2}/2$ from $P_{ij2}$.

A balanced way to extend this approach to the case of multiple correlation is by successively subtracting from $P_{ij}$ an amount proportional to the correlation with each positively correlated criterion. This results in determining the $C_{ij}$, replacing the simple sum by a weighted average

$$C_{iJ} = \sum_{j \in J} W_j P_{ij}, \tag{4}$$

with

$$W_j = \Pi_{j2 \in S-\{j\}}(1 - (\max(S_{jj2}, 0)/2)). \tag{5}$$

### 2.3. The Treatment of Interaction

In a second step, to deal with the possibility of interaction between criteria, the principle of preferences concentration is applied. While collinearity occurs when a common factor is present in two criteria leading to their values moving together, interaction occurs when such a factor amplifies or reduces the joint influence of the two criteria. Applying the principle of concentration of preferences, if two criteria together give a high preference to an alternative, even if it is not the most preferred alternative by each one of them, the high joint preference for that alternative leads us to assign a high capacity to the union of the two criteria.

The occurrence of interaction can be extended from pairs of isolated criteria to pairs of sets with larger numbers of criteria. The interaction between two sets is measured by the difference between the highest preference according to each set separately and according to the two sets together.

More precisely, the capacity of a subset $J$ of the set $S$ of $m$ criteria used in the evaluation of the set of alternatives $\{A_1, \dots, A_n\}$ is determined through the following steps:

First, determine the maximum of the preferences $C_{iJ}$ derived from the treatment of collinearity:

$$M_J = \max_{i \in \{1, \dots, n\}} C_{iJ}. \tag{6}$$

The capacity of the set of criteria $\{C_{j1}, \dots, C_{js}\}$ will be

$$C(J) = M_J / M_S. \tag{7}$$

A numerical assessment of the effect of collinearity and interaction is offered by the distribution of Shapley values [34] of the capacity.

For the criterion $c$ of the set of criteria $S$ and the capacity $\mu$ on $S$, the Shapley value is

$$\text{Shapley}_\mu(c) = \sum_{K \subset S \setminus \{c\}} ((\#(S \setminus \{c\} \setminus K))!(\#(K))!/(\#(S))!)(\mu(K \cup \{c\}) - \mu(K)) \tag{8}$$

If $\mu$ is a probability, that means, if additivity holds, $K$ and $\{c\}$ being disjoints,

$$\mu(K \cup \{c\}) - \mu(K) = \mu(\{c\}) \tag{9}$$

so that, in this case, the Shapley value of $c$ equals the probability.

The Choquet integral of a function $z$ defined on a set $S = \{C_1, \dots, C_m\}$ with respect to a capacity $\mu$ on $S$ is

$$I_\mu(z) = \sum_{j=1}^{m} \left[ z\left(C_{p(j)}\right) - z\left(C_{p(j-1)}\right) \right] \mu\left(\left\{C_{p(j)}, \dots, C_{p(m)}\right\}\right) \tag{10}$$

for $p$, a permutation of $\{1, \dots, m\}$, such that

$$z(C_{p(1)}) \le \cdots \le z(C_{p(m)}) \text{ and } z(C_{p(0)}) = 0 \tag{11}$$

Just to illustrate the concept, it may be interesting to notice that this is equivalent to

$$I_\mu(z) = \sum_{j=1}^{m} z\left(C_{p(j)}\right)\left[\mu(r(p(j))) - \mu(r(p(j+1)))\right] \tag{12}$$

for

$$r(j) = \{j, \dots, m\}, \text{ for all j from 1 to m, and } r(m+1) = 0. \tag{13}$$

To concretize the procedure, let us consider an alternative evaluated by three criteria $C_1$, $C_2$, and $C_3$, with evaluations 90, 70, and 10, respectively. Suppose individual capacities are $\mu(C_1) = 0.3$, $\mu(C_2) = 0.1$, and $\mu(C_3) = 0.6$. In the absence of interaction, the joint evaluation of the alternative by the three criteria would be given by the mean, equal to 40. Let us assume now a strong positive interaction between the criteria $C_1$ and $C_2$ leading to $\mu(\{C_1, C_2\}) = 0.9$. Since these criteria offer high evaluations for the alternative, by the Choquet integral, the joint evaluation $(10 + 0.9(70 - 10) + 0.3(90 - 70) = 70)$ is much higher than the mean. If, on the other hand, a strong negative interaction leads, for example, to $\mu(\{C_1, C_2\}) = 0.3$, the joint evaluation falls to $10 + 0.3(70 - 10) + 0.3(90 - 70) = 34$, smaller than the mean.

The computations involved in the application of these rules in this study employ R [35] and, in particular, [36]. A detailed numerical example is developed in Section 3. Its whole computation is archived in the dataset [37].

## 3. Results and Discussion

### 3.1. Multicriteria Formulation of Negotiation Terms

The multicriteria model is formally defined by the criteria, alternatives, capacity generation algorithm, and composition rule.

There is no limit on the number of criteria. Regarding the criteria, the relevant feature of the rewarded collaboration multicriteria model is the presence of two types of criteria. The first type is related to the motivation of each defendant to reduce the expected sentence. Each of these criteria is based on a single attribute, namely, the number of years in prison—each defendant wants to minimize that number.

The second type is related to the costs and benefits to the criminal justice system. The first criterion of this type is related to the change in the cost of the investigation related to checking and using the information provided in the collaboration proposal. It is measured by the difference between the cost of proceeding against all, not using the information offered, and the cost of checking the piece of information offered and proceeding to obtain sufficient evidence against the non-collaborative suspects.

Other criteria of this second type may be included in the model. One such criterion is the relevance of the proposed information, if it is made available, to dismantling a criminal organization. The evaluation according to this criterion may be given by a classification within a set of distinct levels: null, low, moderate, and high, for instance.

The second element, the set of alternatives, is formed by designating one alternative to each proposal and one to no proposal. The proposals may originate from isolated defendants or from groups of defendants. The alternative of no proposal is represented by the expected values of the attributes that constitute each criterion in the event that no collaboration is accepted.

The capacity generation algorithm starts with the homogenization of the vectors of the assessments of the alternatives by the criteria. This is carried out by the probabilistic transformation into vectors of probabilities of each alternative being the best. The capacity of each set of criteria is then obtained by the maximization of the joint probabilities along the alternatives. An important motive for the collaborators—confidence—comes from this feature of automatic assignment of importance to the criteria; i.e., the relative importance of each criterion is determined solely based on the proper values that it assigns to the alternatives.

Finally, the composition rule assigns to each alternative the Choquet integral of its vector of probabilities of minimization with respect to the above-described capacity. The alternatives are ranked according to these scores.

A step forward is made in conflict resolution when it is possible to consider, together with the objectives of the isolated agents imposing their own wills, the implicit will of the entire collective of negotiators to reach a satisfactory agreement. A legitimate agreement is achieved when the decision power is transferred from the isolated individuals to the collective, subjecting the individuals' wills to the general will.

To confirm the legitimacy of the general will, the agents must recognize in it a clear intent to maximize the satisfaction of the real needs of the parties. This intent is emphasized by evaluating alternatives represented by the preferences of each individual and by assigning importance to the sets of criteria according to the highest level of satisfaction that they provide to one or more negotiators.

Even knowing that a judge's human evaluation will determine the fairness of the negotiation, the transparency granted by the application of the numerical composition rule may be a decisive factor in making these collaborations part of the system and in making both the process of collaboration and the reward for collaboration motivations for collaborators. This methodology may be the decisive ingredient in making rewarded collaboration an important tool in the fight against crime.

### 3.2. An Example

In this section, an example of the application of the methodology is provided. This example clarifies the role of the defendants' initiative in the plea-bargaining process as well as the incentive for rewarded collaboration that the objectivity of the model can contribute to generating.

Table 1 depicts a formulation with three defendants. The possibilities of no collaboration proposal presentation, presentations of individual proposals by each defendant, presentations of proposals by pairs of defendants, and presentation of a joint proposal by all defendants give rise to eight possible alternatives.

**Table 1.** Identification of eight alternatives by five criteria.

|        | [Crit1] | [Crit2] | [Crit3] | [Crit4] | [Crit5] |
|--------|---------|---------|---------|---------|---------|
| [Alt1] | 4       | 4       | 4       | 0       | 2       |
| [Alt2] | 6       | 6       | 0       | 1       | 1       |
| [Alt3] | 6       | 0       | 6       | 1       | 1       |
| [Alt4] | 0       | 6       | 6       | 1       | 1       |
| [Alt5] | 12      | 0       | 0       | 2       | 1       |
| [Alt6] | 0       | 12      | 0       | 2       | 1       |
| [Alt7] | 0       | 0       | 12      | 2       | 0       |
| [Alt8] | 2       | 2       | 2       | 3       | 0       |

A maximum prison sentence of twelve years is used to define the evaluation of the alternatives according to the criteria of penalty length (Crit1, Crit2. and Crit3). This is the punishment for the defendant not participating in a proposal made by the other two. In each other alternative, the importance of nonparticipating defendants' involvement in the crime is estimated by dividing this penalty by their number. Thus, for the alternatives with a single proponent, the penalty for the other two is six years, and, in the case of no proposal presentation, each defendant is supposed to be sentenced to four years. In the final instance, the proponents of the three defendants' proposal accept a two-year sentence for each of them.

Two criteria are used, measuring the cost to the prosecution added by accepting each proposal (Crit4) and the relevance of the information provided by the acceptance of each proposal to the dismantlement of criminal organizations (Crit5). The prosecution's workload is assumed to increase linearly with the number of collaboration proponents. For the provoked damages for the criminal organization involved in the case, the expected loss of power brought to it by using the information corresponding to each alternative is categorized into three levels: high, moderate, and null. It is hypothesized that there is

moderate damage in alternatives 2 to 6 and high damage in alternatives 7 and 8. Naturally, from Alt1, no damage is expected.

Table 2 displays the rank correlation coefficients between the columns of Table 1. Crit1 and Crit2 present a positive correlation with Crit5. The other correlations are all negative.

**Table 2.** Spearman correlation coefficients between the criteria.

|  | [Crit2] | [Crit3] | [Crit4] | [Crit5] |
|---|---|---|---|---|
| [Crit1] | −0.424 | −0.424 | −0.206 | 0.340 |
| [Crit2] |  | −0.424 | −0.206 | 0.340 |
| [Crit3] |  |  | −0.206 | −0.255 |
| [Crit4] |  |  |  | −0.809 |

The probabilities of preference for each alternative according to the isolated criteria are displayed in Table 3. They are derived from the pairwise comparisons of the alternatives' evaluations. Since the criteria were designed with this orientation, the evaluation is counted as better if smaller.

**Table 3.** Probabilities of preference by isolated criteria.

|  | [Crit1] | [Crit2] | [Crit3] | [Crit4] | [Crit5] |
|---|---|---|---|---|---|
| [Alt1] | 0.107 | 0.107 | 0.107 | 0.250 | 0.000 |
| [Alt2] | 0.054 | 0.054 | 0.214 | 0.179 | 0.107 |
| [Alt3] | 0.054 | 0.214 | 0.054 | 0.179 | 0.107 |
| [Alt4] | 0.214 | 0.054 | 0.054 | 0.179 | 0.107 |
| [Alt5] | 0.000 | 0.214 | 0.214 | 0.071 | 0.107 |
| [Alt6] | 0.214 | 0.000 | 0.214 | 0.071 | 0.107 |
| [Alt7] | 0.214 | 0.214 | 0.000 | 0.071 | 0.232 |
| [Alt8] | 0.143 | 0.143 | 0.143 | 0.000 | 0.232 |

To account for collinearity, reduction factors of $1 - 0.34/2 = 0.83$ are applied to the probabilities of preference according to Crit1 and Crit2 and of $(1 - 0.34/2)2 = 0.69$ to the probabilities of preference according to Crit5. To illustrate the effect of this reduction procedure, the derivation of the capacities for Crit4 and Crit5 and for the pairs of criteria including Crit5 is demonstrated in Table 4.

**Table 4.** Example of capacities determination.

|  | [Crit4] | [Crit5] | [1&5] | [2&5] | [3&5] | [4&5] |
|---|---|---|---|---|---|---|
| [Alt1] | 0.250 | 0.000 | 0.089 | 0.089 | 0.107 | 0.250 |
| [Alt2] | 0.179 | 0.074 | 0.118 | 0.118 | 0.288 | 0.252 |
| [Alt3] | 0.179 | 0.074 | 0.118 | 0.118 | 0.127 | 0.252 |
| [Alt4] | 0.179 | 0.074 | 0.252 | 0.252 | 0.127 | 0.252 |
| [Alt5] | 0.071 | 0.074 | 0.074 | 0.074 | 0.288 | 0.145 |
| [Alt6] | 0.071 | 0.074 | 0.252 | 0.252 | 0.288 | 0.145 |
| [Alt7] | 0.071 | 0.160 | 0.338 | 0.338 | 0.16 | 0.231 |
| [Alt8] | 0.000 | 0.160 | 0.278 | 0.278 | 0.303 | 0.160 |
| Maximum | 0.250 | 0.160 | 0.338 | 0.338 | 0.303 | 0.252 |
| Capacity | 0.426 | 0.272 | 0.575 | 0.575 | 0.516 | 0.430 |

Disregarding the possible presence of interaction, the application of the principle of preferences concentration leads to a weighted average composition with weights proportional to the maxima of the preference vectors, or, alternatively, to the maxima of the preference vectors corrected to take into account collinearity. The scores can then be made to sum 1 and interpreted as global probabilities of preference.

The capacities for the composition considering interaction can also be built on two different bases. They can be derived directly from the preference vectors or from the matrix obtaining the reduction of the vectors of preference by the criteria with a positive correlation with other criteria.

The comparative analysis of the proposals can be developed by examining the results of the application of these four different automatic composition rules. Table 5 provides information on the effects of the different assumptions. The weights obtained disregarding interaction are presented in the first rows of Table 5, the first directly derived from the probabilities of preference and the second derived from reduced vectors for the criteria with collinearity. After these rows of probabilistic weights are the Shapley values for the capacities derived from the matrices of probabilities of preference and from the matrices with reduced columns for the criteria with collinearity.

**Table 5.** Probabilistic weights and Shapley values.

|  | [Crit1] | [Crit2] | [Crit3] | [Crit4] | [Crit5] |
|---|---|---|---|---|---|
| Probability | 0.190 | 0.190 | 0.190 | 0.222 | 0.206 |
| Probability & collinearity | 0.181 | 0.181 | 0.219 | 0.255 | 0.163 |
| Choquet | 0.218 | 0.218 | 0.151 | 0.176 | 0.237 |
| Choquet & collinearity | 0.201 | 0.201 | 0.204 | 0.217 | 0.176 |

Table 5 illustrates how Crit5 loses importance when collinearity is considered. The second higher effect of collinearity correction is an increase in the importance of Crit4. The interaction correction affects these two criteria in the opposite direction, albeit to a lesser degree.

The scores obtained with and without collinearity and interaction modifications are displayed in Table 6. The joint proposal of the first two defendants, which results in significant harm to the criminal organization, is ranked first, with a score of 0.202 for the complete treatment of collinearity and interaction, the other scores varying between 1.47 and 1.58. Its score is also significantly higher than the others for all other composition rules.

**Table 6.** Scores for various composition rules.

|  | Probabilistic | Probabilistic & Collinearity | Choquet | Choquet & Collinearity |
|---|---|---|---|---|
| [Alt1] | 0.117 | 0.126 | 0.132 | 0.158 |
| [Alt2] | 0.123 | 0.129 | 0.139 | 0.157 |
| [Alt3] | 0.123 | 0.123 | 0.140 | 0.147 |
| [Alt4] | 0.123 | 0.123 | 0.140 | 0.147 |
| [Alt5] | 0.120 | 0.121 | 0.148 | 0.165 |
| [Alt6] | 0.120 | 0.121 | 0.148 | 0.165 |
| [Alt7] | 0.145 | 0.134 | 0.206 | 0.202 |
| [Alt8] | 0.130 | 0.121 | 0.162 | 0.156 |

In contrast, Alt1 and Alt8 have significant rank variation if collinearity or interaction is neglected. Alt8 is ranked second if the importance of Crit5 is not reduced by the correction for collinearity and last in the weighted average composition with the correction for collinearity. Alt1 is the last if the collinearity correction is not applied and is in an intermediary position if it is.

## 4. Conclusions

This article examines a tool designed to improve the practice of plea bargaining. It promotes collaboration by using a multicriteria model to rank collaboration proposals.

In addition to the explicit interests of the parties, defendants, and prosecution, the model implicitly incorporates society's interest in co-operation into the negotiation pro-

cess. An automatic composition rule applies the general will by evaluating alternative representations based on a comprehensive set of criteria.

The model contributes value to the collaborations, thereby attracting collaborators. Their participation may transform rewarded collaboration into a tool for reducing principal-agent inefficiencies in criminal punishment. Promoting competition among collaboration proposals in an objective basis may produce a cultural shift, effectively contributing to the strengthening of collaboration, the dilution of criminal organizations, and the diminution of incarceration.

The model identifies objectively the alternatives that maximize the preferences of negotiation participants. Concurrently, it permits the consideration of different ways to combine the measures of importance of the criteria, taking into account their collinearity and interaction.

The preference quantification is based on counting procedures and automated criteria valuation. A new element of the preferences combination algorithm is a simple rule for adjusting the importance of positively correlated criteria. Different situations are modeled in an illustrative example, with various alternatives. Variations in asymmetry between the penalties are considered, as well as between the prosecution criteria.

Taking collinearity into account, the proposed ranking methodology may result in an increase in the feasible number of criteria, which raises possibilities that should be addressed in future developments. It is interesting to determine, for example, to what extent this increase in the number of criteria enables the automatic capacities derivation mechanism to implement the principle of preferences concentration more faithfully. The effectiveness of the proposed method may be demonstrated by comparing the outcomes of applying to actual situations models with progressively more criteria.

**Author Contributions:** Conceptualization, A.P.S., L.O.G. and T.L.S.; methodology, A.P.S., L.O.G. and T.L.S.; software, A.P.S., L.O.G. and T.L.S.; validation, A.P.S., L.O.G. and T.L.S.; formal analysis, A.P.S., L.O.G. and T.L.S.; investigation, A.P.S., L.O.G. and T.L.S.; resources, A.P.S., L.O.G. and T.L.S.; data curation, A.P.S., L.O.G. and T.L.S.; writing—original draft preparation, A.P.S., L.O.G. and T.L.S.; writing—review and editing, A.P.S., L.O.G. and T.L.S.; visualization, A.P.S., L.O.G. and T.L.S.; supervision, A.P.S., L.O.G. and T.L.S.; project administration, A.P.S., L.O.G. and T.L.S.; funding acquisition, A.P.S., L.O.G. and T.L.S. All authors have read and agreed to the published version of the manuscript.

**Funding:** This research received no external funding.

**Data Availability Statement:** The data published in this study are openly available in Sant'Anna, Prisioneiros, Mendeley Data V4, online at https://doi.org/10.17632/7d5f7v6x3g.4.

**Conflicts of Interest:** The authors declare no conflict of interest.

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
