# Peer review of "A Multicriteria Standard to Rank Plea Bargain Proposals"

_standards, doi:10.3390/standards3020016_

Round 1

Reviewer 1 Report

Comments and Suggestions for Authors

Dear Authors

The topic is interesting and innovative. The contribution to the discussion of the topic is significant. Below I present some suggestions for adjustment to the current version:

1) Insert at the end of the introduction a short paragraph summarizing the whole structure of the paper;

2) Equations should be centered and numbered to the right;

3) The list of references is in disagreement with the authors' instructions; read and review the references;

4) Tables 4 and 6 are not formatted according to American standards;

5) I suggest a more detailed conclusion.

Best Regards

Reviewer

Author Response

Point 1: Insert at the end of the introduction a short paragraph summarizing the whole structure of the paper.

 Response 1: A paragraph was added at the end of the Introduction to briefly describe the paper's structure.

Point 2: Equations should be centered and numbered to the right.

Response 2: Equations are now aligned in the center and numbered according to the template.

Point 3: The list of references is in disagreement with the authors' instructions; read and review the references.

 Response 3: The presentation of the references has been corrected.

Point 4: Tables 4 and 6 are not formatted according to American standards.

Response 4: Separation points are now used in the presentation of the numbers in the final column of Table 4 and in the first column of Table 6.

Point 5: I suggest a more detailed conclusion.

Response 5: The conclusion section was expanded to include suggestions for future developments, with a focus on the importance of practical applications to provide evaluation of the possibility of increasing the number of criteria made possible by the proposed collinearity treatment.

Reviewer 2 Report

Comments and Suggestions for Authors

Dear authors:

I begin with congratulations for the subject studied in the article as well as for the methodology used. I think it is a great contribution of multicriteria decision techniques applied to a field in which they are not commonly used, the world of justice.

However, there are some comments that I would like to make for a better understanding of the content of the article.

1.- The “Materials and Methods” section should be devoted exclusively to the multicriteria methodology used. To this end:

              1.1.- Lines 153 to 212 should be moved to the “Introduction” part.

              1.2.- Begin “Materials and Methods” with a brief summary of the basic method used (CPP) and the corrections for collinearity and interaction.

              1.3.- The “Materials and Methods” section should consist of 3 separate parts: the first part dedicated to the exposition of the CPP, the second part dedicated to the treatment of collinearity and the third part dedicated to the treatment of interaction.

The treatment of the interaction between criteria using the Choquet integral should be described in more detail. References are mentioned, but the description is very brief, which complicates understanding.

I trust that these comments will serve to improve the good work already done.

Author Response

Point 1: The “Materials and Methods” section should be devoted exclusively to the multicriteria methodology used. To this end:

Lines 153 to 212 should be moved to the “Introduction” part.

Response 1: We acted as suggested. We believe that, in addition to giving prominence to the multicriteria methodology, moving the details of the current system of rewarded collaboration to the Introduction made it easier for the reader to comprehend the practical significance of the application.

Point 2: Begin “Materials and Methods” with a brief summary of the basic method used (CPP) and the corrections for collinearity and interaction.

Response 2: An initial paragraph was added to the referenced section providing the suggested information.

Point 3: The “Materials and Methods” section should consist of 3 separate parts: the first part dedicated to the exposition of the CPP, the second part dedicated to the treatment of collinearity and the third part dedicated to the treatment of interaction.

Response 3: The referenced section now contains the three distinct subsections indicated. Specific information was added to the description of the treatment of collinearity and interaction, two important features of the proposed approach.

Point 4: The treatment of the interaction between criteria using the Choquet integral should be described in more detail. References are mentioned, but the description is very brief, which complicates understanding.

Response 4: A different formulation of the Choquet integral and a numerical illustration of its computation have been added.

Round 2

Reviewer 1 Report

Comments and Suggestions for Authors

Dear Authors

Initially, I congratulate you for the extensive revision you have done and for implementing the suggestions indicated by the reviewers in the first round of revision. The current version is fit for publication.

Best Regards

Reviewer